# Self-probing spectroscopy of XUV photo-ionization dynamics in atoms subjected to a strong-field environment

Doron Azoury[1], Michael Krüger [1], Gal Orenstein[1], Henrik R. Larsson [2], Sebastian Bauch[3], Barry D. Bruner[1] & Nirit Dudovich[1]

Single-photon ionization is one of the most fundamental light matter interactions in nature, serving as a universal probe of the quantum state of matter. By probing the emitted electron, one can decode the full dynamics of the interaction. When photo-ionization is evolving in the presence of a strong laser field, the fundamental properties of the mechanism can be signicantly altered. Here we demonstrate how the liberated electron can perform a self-probing measurement of such interaction with attosecond precision. Extreme ultraviolet attosecond pulses initiate an electron wavepacket by photo-ionization, a strong infrared field controls its motion, and finally electron–ion collision maps it into re-emission of attosecond radiation bursts. Our measurements resolve the internal clock provided by the self-probing mechanism, obtaining a direct insight into the build-up of photo-ionization in the presence of the strong laser field.

[1] Department of Physics of Complex Systems, Weizmann Institute of Science, Rehovot 76100, Israel. [2] Institut für Physikalische Chemie, Christian-Albrechts-Universität zu Kiel, D-24098 Kiel, Germany. [3] Institut für Theoretische Physik und Astrophysik, Christian-Albrechts-Universität zu Kiel, D-24098 Kiel, Germany. Azoury D and Krüger M contributed equally to this work. Correspondence and requests for materials should be addressed to N.D. (email: nirit.dudovich@weizmann.ac.il)

Ultrafast spectroscopy has advanced significantly during the past decades as a fundamental tool that manipulates and probes the temporal evolution of a quantum system. One of the most important breakthroughs in ultrafast science was achieved with the production of attosecond ($10^{-18}$ s) laser pulses in the extreme ultraviolet (XUV) wavelength range via a process known as high harmonic generation (HHG)[1,2]. These developments opened the door to a new time regime that had previously been considered inaccessible, allowing one to follow multi-electron dynamics in atoms[3], molecules[4], surfaces,[5] and solids[6].

The high nonlinearity that underlies the attosecond pulse generation process offers a unique path for ultrafast measurements. This spectroscopic approach, known as attosecond self-probing or HHG spectroscopy, exploits a built-in pump–probe process driven by an intense infrared (IR) laser field every half-cycle of its oscillation[7]. Here strong-field tunneling ionization acts as a pump, removing an electron and creating a hole in the system. Driven by the laser field, the liberated electron wavepacket returns to the parent ion and probes the hole via radiative recombination, which leads to the emission of high-order harmonics of the driving laser field (Fig. 1a). This nonlinear parametric process, which starts and ends at the ground state of the system, serves as an internal clock, encoding the evolution of the system between ionization and recollision with attosecond precision. In addition, this scheme provides a high spatial resolution

since the de-Broglie wavelength associated with the recolliding electron wavepacket is on an Ångström length scale[8]. The combination of extremely high temporal and spatial accuracy allows the observation of a range of fundamental phenomena—for example, proton motion[9], valence orbital hole and electron dynamics in molecules[4], hole rotation in chiral molecules[10], and tunneling[11].

Although attosecond self-probing holds great promise for ultrafast spectroscopy and control of matter, the main limitations are imposed by its starting point—strong-field tunneling ionization. Since the tunneling probability decays exponentially with electron binding energy, it allows probing of only a narrow range of valence shell orbitals. Moreover, in systems with multiple orbitals, the tunneling mechanism dictates their relative amplitudes and phases. Finally, tunneling is temporally constrained to the peak of the optical field—imposing a major limitation on the ability to manipulate the temporal properties of the dynamics under study.

An alternative approach to generate high harmonics applies a photo-ionization process as the initial step of the recollision mechanism. This approach is called XUV-initiated HHG,[12,13] which decouples the ionization step from the subsequent steps of the interaction by replacing strong-field-induced tunnel ionization with photo-ionization driven by an attosecond XUV pulse, as schematically shown in Fig. 1b. Photo-ionization serves as a

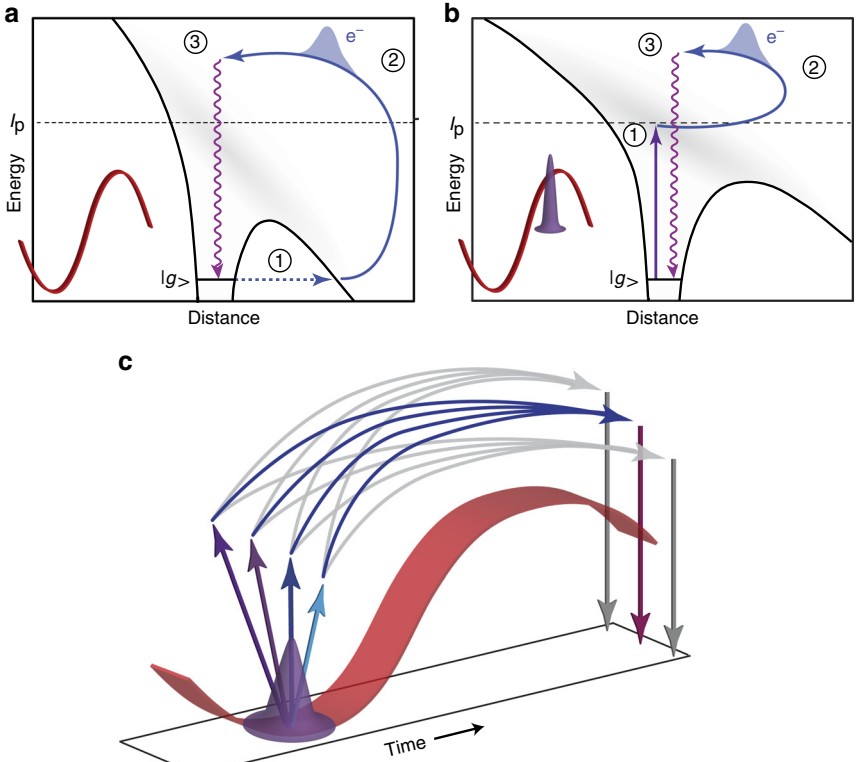

**Fig. 1** Photo-ionization self-probing spectroscopy. **a** Schematic diagram of tunneling-initiated high harmonic generation (HHG). The first step is tunnel ionization from the ground state $|g\rangle$ through a suppressed Coulomb barrier (solid black line) induced by a strong laser field (illustrated in red). **b** Schematic diagram of extreme ultraviolet (XUV)-initiated HHG. In addition to the strong laser field, the atom is driven by a synchronized XUV attosecond pulse (illustrated in purple) that creates an electron wavepacket through direct photo-ionization (step 1). In both schemes, blue arrows represent strong-field-driven electron trajectories (step 2) that terminate in recollision with the parent ion (step 3), leading to the emission of high harmonics. **c** Interferometric picture of the XUV-initiated HHG mechanism. The photo-ionization attosecond pulse (purple) creates an electron wavepacket through several quantum paths (arrows pointing up), acting as the starting point of each arm of the interferometer. The infrared (IR) field (red) accelerates the wavepacket and defines the phase evolution of the paths through strong-field electron trajectories (blue and gray arrows). All quantum paths leading up to the same recollision energy interfere, determining the intensity of the corresponding harmonic emission (arrows pointing down). The XUV-IR delay controls which paths will interfere constructively (blue arrows) or destructively (gray arrows). Measurements of the intensity of the new harmonics as a function of XUV-IR delay allow the reconstruction of the photo-ionization dynamics

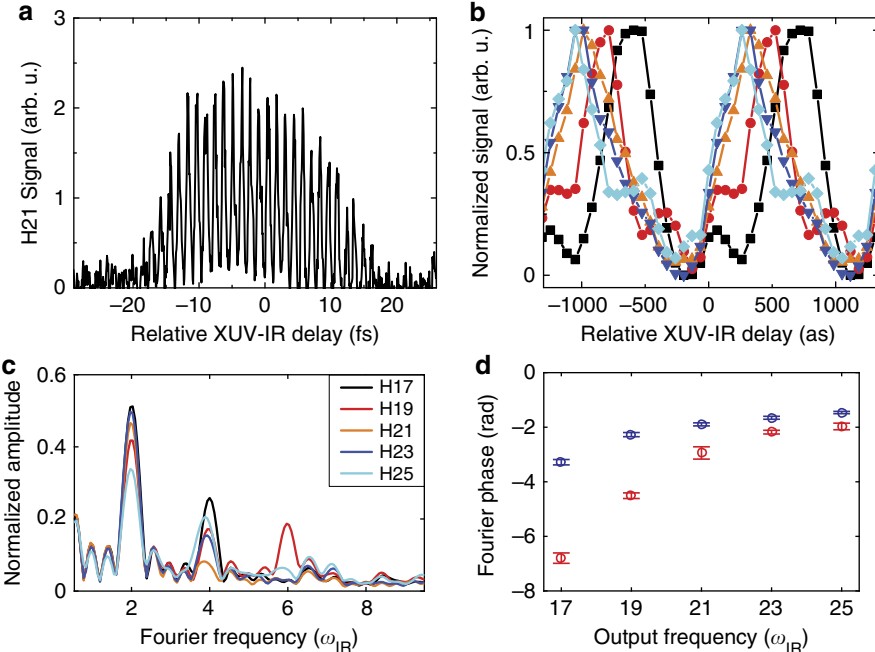

**Fig. 2** Spectral analysis of multi-quantum-path beating in XUV-initiated HHG. **a** Intensity of harmonic 21 as function of the relative XUV-IR delay. Zero delay is defined arbitrarily. **b** Intensity of harmonics 17–25 as a function of the relative XUV-IR delay. All harmonics oscillate with a dominant frequency of $2\omega_{IR}$ and peak at different delays in a descending order. For better visibility, the signal is integrated over seven oscillation cycles, normalized and displayed over several cycles. The different harmonics are represented by colors as in **c**. **c** Fourier analysis of the oscillating signal of harmonics 17–25. Strong anharmonic components at $4\omega_{IR}$ are visible, as well as signatures of $6\omega_{IR}$ oscillations. **d** Measured Fourier phases (circles) as a function of harmonic order at Fourier frequencies of $2\omega_{IR}$ and $4\omega_{IR}$, in blue and red, respectively (error bars: standard deviation; for details, refer to Methods section)

universal probe of a large range of quantum systems. Ionization by single photons accurately probes, in a linear manner, the quantum state of the matter under scrutiny and produces photo-electrons, which are commonly measured far away and long after the interaction. Biegert et al.[14] performed a pioneering demonstration of XUV-initiated HHG showing that the HHG mechanism can be triggered by an external XUV field. Brizuela et al.[15] showed that low-order harmonics can lead to enhancement of the HHG yield, and it has been suggested that XUV-initiated HHG contributes to the gain in other HHG enhancement experiments[16,17]. Gademann et al.[18] demonstrated the ability to manipulate the temporal and spectral properties of the HHG process by controlling the XUV-IR delay (see also related work with photoelectrons[19,20]). The establishment of the XUV-initiated HHG mechanism as a new spectroscopic approach requires a significant step forward. XUV-initiated HHG is a highly nonlinear phenomenon that couples XUV and IR frequencies to generate a new set of XUV frequencies. Its application as a spectroscopic scheme raises the following questions: Can we describe the strong-field interaction via recollision trajectories, as required for a self-probing approach? What is the spectroscopic insight provided by XUV-initiated HHG?

In this paper, we integrate single-photon ionization—one of the most fundamental light–matter interactions—with self-probing spectroscopy, demonstrating a unique spectroscopic approach in attosecond science. We combine the universality provided by single-photon ionization with the high resolution provided by the recollision self-probing mechanism. Our study reveals the rich, multiple quantum path nature of the underlying mechanism, demonstrating that it opens a route in attosecond time-resolved spectroscopy. The establishment of photo-ionization self-probing spectroscopy is based on several fundamental steps. First, we identify and control the main quantum paths that contribute to the process—each path is initiated by the

incoming XUV field and coupled to the other paths by the strong-field interaction. Next, we perform an independent measurement of the primary steps of the interaction. By applying a weak perturbative field, we independently probe either the XUV excitation step or the strong-field interaction. The measurement identifies the underlying dynamics associated with each step in XUV-initiated HHG. The complete and detailed understanding of the interplay between the XUV excitation and the strong-field interaction enables us to proceed into the final stage of our spectroscopic study. Here we reconstruct both the amplitude and phase associated with each XUV excitation path. The coherent superposition of these paths represents the temporal build-up of the photo-ionized wavepacket at its origin.

## Results

**Photo-ionization self-probing spectroscopy**. We demonstrate photo-ionization self-probing spectroscopy in a collinear scheme composed of two main stages (for a detailed description, see Methods section and Supplementary Note 3). In the first stage, an attosecond pulse train (APT) is generated by focusing an intense IR laser pulse into a source gas cell filled with xenon. The IR and APT beams co-propagate and are refocused by a curved two-segment mirror, which controls their relative delay $\Delta t$, into a target gas cell filled with helium in order to produce XUV-initiated HHG. For an IR intensity of $8 \times 10^{13}$ W cm$^{-2}$, tunnel ionization and, hence, direct HHG by the IR field is strongly suppressed in helium, providing a background-free measurement. The APT consists of 4 discrete odd harmonics in the XUV ranging from $11\omega_{IR}$ to $17\omega_{IR}$, where $\omega_{IR}$ is the angular frequency of the IR field. When the APT and the IR pulse are temporally overlapped in the target gas cell, we observe the generation of at least five odd harmonics at photon energies above the field free helium ionization threshold (see Supplementary Fig. 8 and

Supplementary Note 5). We tested the generality of this observation by performing the experiment with neon atoms and found similar results (see Supplementary Fig. 9). Scanning $\Delta t$ leads to high-contrast oscillations of the new harmonics (Fig. 2a, b), at a dominant frequency of $2\omega_{IR}$[18]. Each individual harmonic can be associated with an optimal delay for which the signal is maximized. As can be clearly observed, this delay gradually increases with descending harmonic order over a range of $\sim$ 500 as (see Fig. 2b).

A closer examination of the oscillating signal shows a significant anharmonicity, manifested in higher-order oscillations at frequencies of $4\omega_{IR}$ and $6\omega_{IR}$ (see Fig. 2c, d). These oscillations reveal the interferometric nature of the mechanism (see Fig. 1c). XUV photo-ionization initiates an electron wavepacket that spreads over several quantum paths (denoted as $n$), with initial amplitude and phase $a_n$ and $\phi_n$, respectively. The IR field accelerates the wavepacket and defines the phase evolution of the paths through strong-field electron trajectories. Upon recollision with the parent ion, the quantum paths interfere and lead to the new set of harmonics. Essentially, multiple harmonic frequencies $n\omega_{IR}$ injected by the initial XUV field are projected through the interaction into a set of new harmonic frequencies $N\omega_{IR}$. We manipulate the phase between the different paths in a linear manner by controlling $\Delta t$. Accordingly, an interferogram which represents the emission of one new harmonic can be described as:

$$I_N(\Delta t) \propto \left| \sum_n a_n \cdot exp\left\{ -i\phi_n + i\Theta_{n,N} + in\omega_{IR}\Delta t \right\} \right|^2, \quad (1)$$

where $\Theta_{n,N}$ represents the phase that is associated with the electron trajectory that maps an initial harmonic $n\omega_{IR}$ into a new harmonic of order $N\omega_{IR}$. Scanning $\Delta t$ leads to a beating of the interferogram's signal. Figure 2d shows the phases of the beating as a function of $N\omega_{IR}$. Beating at $2\omega_{IR}$ is associated with the interference between pairs of neighboring paths separated by $2\omega_{IR}$, e.g., those initiated by 13 and $15\omega_{IR}$. Higher-order beating at $4\omega_{IR}$ and $6\omega_{IR}$ is associated with the interference between paths separated by $4\omega_{IR}$ and $6\omega_{IR}$, respectively. It indicates that, in addition to harmonic 17 whose energy is above helium's ionization threshold, harmonics 11–15 have a significant contribution to the photo-ionization mechanism. This observation is striking; in the presence of the strong infrared field, the atom is ionized by a broad spectral range of XUV light well below the ionization threshold and far from any atomic resonance. This is one of the main results of our work and will be investigated in detail further below. The final step of the interferometer encodes the temporal evolution of the ionized electronic wavepacket via the coupling between the ionization frequencies to the new HHG frequencies. In order to decode this information, we need a better understanding of the dynamics associated with each of the interferometer's arms. Can we think here in the language of recollision trajectories? Can we identify their starting point? Answering these questions experimentally requires a deep study of the basic components that comprise the interferometer—ionization and recollision.

**Perturbative study of the mechanism**. We study the interferometer's dynamics by adding an additional degree of freedom, provided by a weak second harmonic (SH) field of the IR pulse. This approach has been previously applied to probe the internal dynamics of the conventional HHG mechanism[21]. The total field is given by $E(t) = E_0^{IR}[\cos(\omega_{IR}t) + \varepsilon \cos(2\omega_{IR}t + \varphi)]$, where $E_0^{IR}$ is the IR field strength, $\varepsilon \ll 1$ is the SH-IR field ratio and $\varphi$ is their relative phase. In the first experiment, we synchronize the SH and the IR pulses in the target gas cell in order to perturb the strong-

field interaction with helium (see Methods section and Supplementary Note 4). The presence of the perturbative SH field breaks the symmetry of the interaction, which manifests itself in the generation of even harmonics[21]. The mechanism for symmetry breaking is based on the perturbation of the electron trajectories in subsequent half-cycles (see Fig. 3a). The perturbation modifies the phase proportional to the semiclassical action $S$, which accumulates as the electron wavepacket interacts with the strong field. Such a perturbation, expressed by a small phase shift $dS$, is complex in general[22,23]. The imaginary component is associated with a small modification of the ionization probability, whereas its real component is associated with the perturbation of electron propagation in the continuum. The intensity $\tilde{I}_N$ of a given harmonic $N$ now depends on $dS$ as

$$\tilde{I}_N = I_N \begin{pmatrix} |\cos(\varepsilon dS(N,\varphi)/\hbar)|^2, & N \, odd \\ |\sin(\varepsilon dS(N,\varphi)/\hbar)|^2, & N \, even, \end{pmatrix} \quad (2)$$

where $I_N$ is the unperturbed harmonic intensity. Since $dS$ depends on all the different steps of the strong-field interaction, it serves as an accurate probe of the electron dynamics associated with each trajectory.

Applying the perturbative scheme, we measured the HHG spectrum as a function of both the XUV-IR and SH-IR delays. The intensity of each harmonic oscillates with half periodicity of the IR field in both degrees of freedom. The optimal XUV-IR delay of each harmonic order is equivalent to the one-dimensional results. Following the harmonic signal as a function of the SH-IR phase, we clearly observe out-of-phase oscillations of the even and odd harmonics (see Fig. 3b). According to Eq. 2, our results indicate that $dS$ is predominantly real, therefore the mechanism is not dominated by field-induced tunneling. A closer examination shows that the SH-IR phase, which is associated with both even and odd harmonics, slowly drifts with the harmonic order (Fig. 3c). Such response reflects the variation of the electron trajectory length, providing a direct evidence for electron trajectories as the fundamental mechanism of the strong-field interaction. Following the excitation by the XUV field, electrons face the atomic potential barrier modified by the IR laser field. The instantaneous shape of the barrier, and therefore the ionization mechanism itself, changes on a sub-optical-cycle time scale. Here the internal clock provided by the self-probing scheme plays an important role; it provides a direct mapping between the ionization time and the emitted energy, enabling us to probe the evolution of this mechanism within the optical cycle. Specifically, our experiment is able to distinguish if electrons undergo tunneling or over-the-barrier emission (OBE) when launched on their recollision trajectories at a specific time within the cycle. In contrast to OBE, tunneling will lead to a non-vanishing imaginary $dS$, which in turn will cause a deviation from the out-of-phase behavior and strong correlation between XUV-IR and SH-IR oscillations. Neither of these signatures are observed for the XUV-initiated harmonics in our experiment. Ionization by XUV light creates a regime where HHG is dominated by OBE, an effect previously only observed in IR-driven below-threshold HHG[24,25].

A comprehensive understanding of the spectroscopic scheme requires an independent study of the ionization mechanism. To this end, we synchronize the SH and IR pulses in the source gas cell, allowing a direct control over the spectral components of the ionizing APT (see Methods section and Supplementary Note 4). The ionizing attosecond pulses, generated by the SH-IR field, are composed of both even and odd harmonics. We control their balance by manipulating the SH-IR delay according to Eq. 2 (Fig. 3d). Initiating ionization with even harmonics opens up new

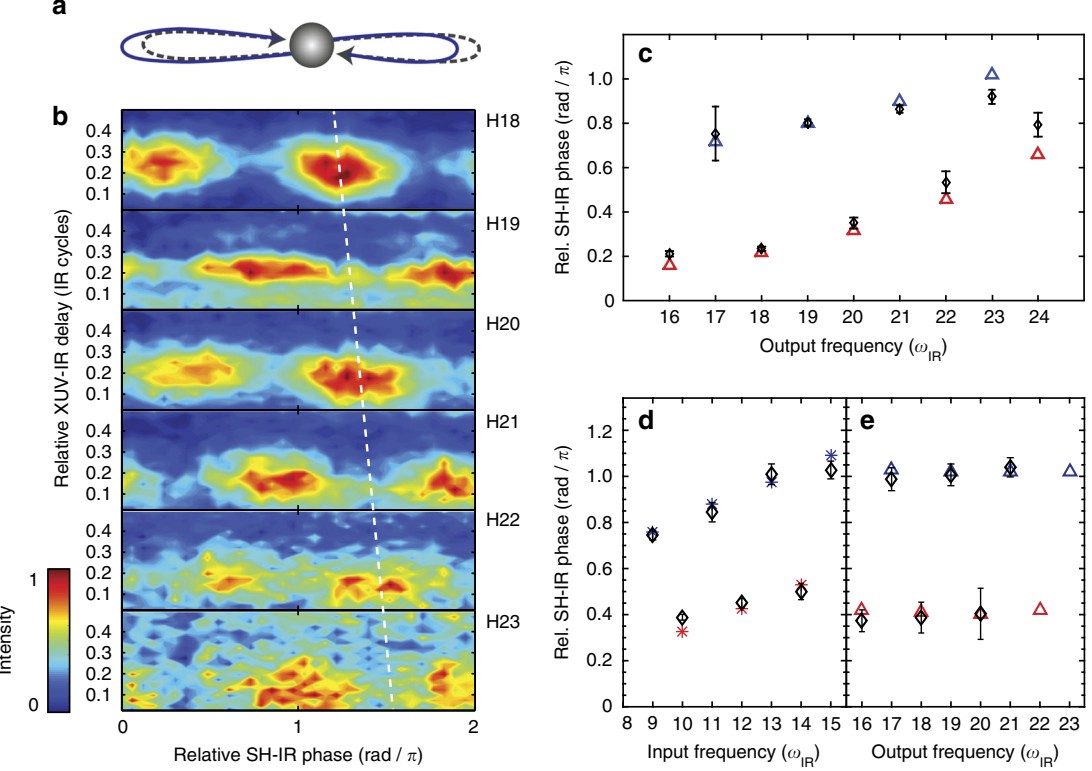

**Fig. 3** Perturbative probing of the XUV-initiated HHG dynamics. **a** Illustration of symmetry breaking between consecutive electron trajectories. For a specific SH-IR delay, the symmetric picture (dashed line) is broken: in one IR half cycle (right), the electron trajectory is shorter than in the consecutive half cycle (left). **b** Probing of the strong-field interaction. Normalized intensity of XUV-initiated harmonics as a function of both SH-IR phase and XUV-IR delay. Here the SH and IR pulses are synchronized in the target gas cell. In addition to XUV-IR delay-dependent oscillations, out-of-phase oscillations between odd and even harmonics are observed as a function of SH-IR phase. The white line is a guide to the eye to highlight the phase slope of the SH-IR oscillations of the even harmonics. **c** SH-IR phases that optimize the intensity of individual harmonics after integrating over the XUV-IR delay, for the experiment (black diamonds) and the Coulomb-corrected model (triangles). **d** Probing of the ionization step. Optimal SH-IR phases of the ionizing harmonics, when the SH and IR fields are synchronized in the source gas cell. The experimental results (black diamonds) show out-of-phase oscillations of odd and even harmonics throughout the entire spectrum. Both the odd and the even phase slopes follow the classical prediction (Eq. 2, stars). **e** Optimal SH-IR phases for XUV-initiated harmonics above the helium ionization threshold. The flat-phase behavior is in good agreement with the Coulomb-corrected model (triangles). All error bars on experimental data correspond to standard deviation

quantum paths that consequentially lead to the generation of XUV-initiated even harmonics. Even and odd XUV-initiated harmonics oscillate out of phase when scanning the SH-IR phase and the oscillation phase of each parity group is uniform as a function of harmonic number (Fig. 3e). These two observations reveal two important properties of the ionization mechanism. First, out-of-phase oscillations indicate that the process retains the SH-IR phase behavior of the ionizing harmonics—odd and even harmonics correspond to two sets of independent quantum paths. Second, the uniform phase response shows that all the quantum paths that contribute to the new harmonics' generation process are initiated by the same ionized electronic wavepacket. These observations raise the following challenge: Can we reconstruct the XUV-initiated electron wavepacket, revealing its attosecond evolution in the presence of the strong IR field?

**Reconstructing the build-up of the electron wavepacket**. Perturbing both the ionization step and the strong-field recollision trajectories resolves the underlying dynamics that defines the different quantum paths in the XUV-initiated HHG process. As we have demonstrated, the ionization is dominated by broadband XUV photo-ionization. The ionized electronic wavepacket populates several independent paths. Finally, the evolution of the wavepacket follows well-established strong-field trajectories[26], mapping its nonlinear dynamics into a new set of HHG

frequencies. These fundamental steps can be integrated into a Coulomb-corrected three-step model (CCTSM; see Methods section and Supplementary Note 1). The CCTSM describes the nonlinear dipole response of an atomic system to a combined XUV-IR field. The model does not assume anything about the XUV excitation or the excited states involved in the mechanism; this allows for an investigation of these dynamics within the framework of self-probing spectroscopy. Electrons are liberated by OBE and uniformly populate strong-field electron trajectories; here the barrier is formed by the combined laser-Coulomb potential[25]. An excellent agreement of the model with the SH-perturbative measurements strongly supports our approach (see Fig. 3c, e). We also performed a numerical integration of the time-dependent Schrödinger equation (TDSE) using the time-dependent generalized-active-space configuration-interaction (TD-GAS-CI) method[27,28] for helium, taking electronic correlation effects into account. The simulation shows XUV-IR oscillations, similar to the experiment. In addition, it allows us to visualize the electron trajectories, confirming our understanding of the mechanism (see Methods section and Supplementary Note 2).

Revealing the primary paths of the nonlinear mechanism provides us with the opportunity to trace the attosecond build-up of the electronic wavepacket, directed by the interplay of the XUV and the strong IR fields' interaction with the atomic system. The

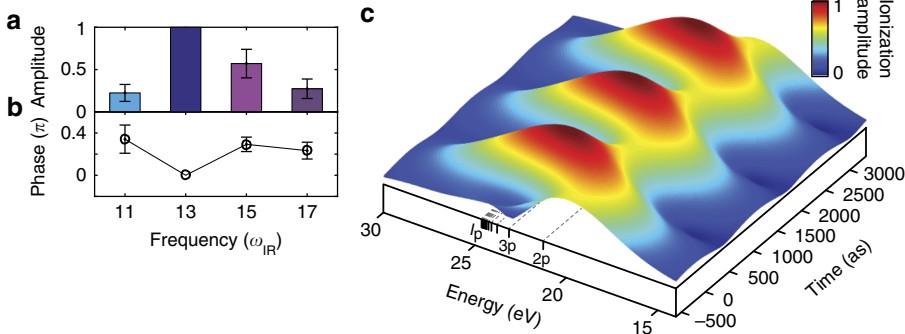

**Fig. 4** Reconstruction of the ionization dynamics in strongly driven helium. Spectral **a** amplitude and **b** phase. We fix the amplitude and phase of the frequency component at $13\omega_{IR}$ since the absolute amplitude and phase of the wavepacket are unknown (error bars: standard deviation). **c** Time–energy (Gabor) representation of the ionized electron wavepacket build-up; ionization amplitude as a function of time and energy. The energy range of the ionization spans from about 9 eV below the field-free ionization potential ($I_p$) of helium to the continuum. We display the field free excited $p$-state manifold of helium for orientation

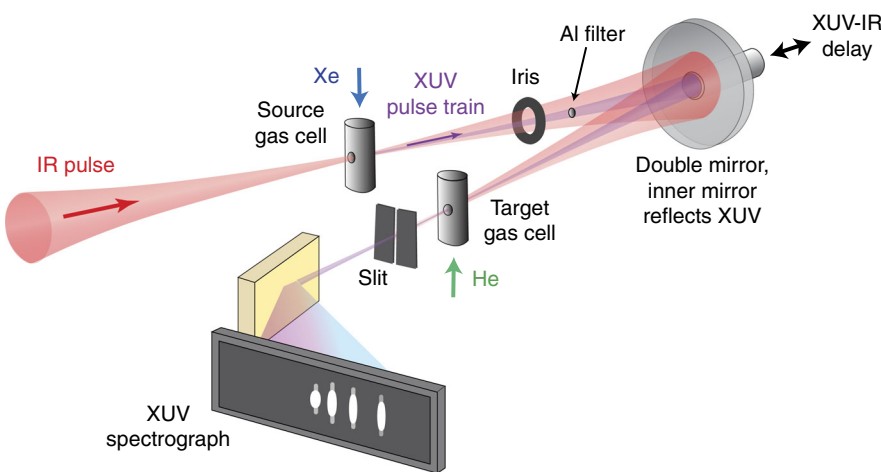

**Fig. 5** Experimental setup. The IR beam is focused into the source gas cell where an attosecond pulse train (APT) is generated. The inner part of the IR beam is filtered out with a thin Al foil in order to obtain spatially independent pump (APT) and probe (IR) beams. The temporal delay between the beams is controlled by moving the inner segment of the concave focusing mirror. Both beams are refocused into a target gas cell, generating XUV-initiated HHG. The spectrum resulting from the interaction is measured in a grating-based XUV spectrograph

spectral components of this wavepacket, defined by $a_n$ and $\phi_n$ (see Eq. 1), are mapped into the observed beating patterns at frequencies of $2\omega_{IR}$, $4\omega_{IR}$ and $6\omega_{IR}$. We decompose the complex contribution of each path by applying a reconstruction procedure to the Fourier data displayed in Fig. 2. In this scheme, the quantum path propagation and interference are evaluated by the CCTSM model while $a_n$ and $\phi_n$ are unknown (see Methods section). These parameters depend on both phase and amplitude of the harmonics that form the ionizing XUV pulses and the transition dipole from the ground state to the excited states. Fig. 4a, b show the reconstructed amplitude and phase, respectively. Figure 4c displays a time–energy (Gabor) representation of the reconstructed ionization dynamics in helium. The ionization bandwidth reaches as deep as 9 eV below the field-free ionization potential of helium and shows an approximately uniform response in both amplitude and phase to the attosecond pulse. These observations indicate the lack of pronounced resonances and hence strongly support the picture of a continuum-like excited state manifold in helium, as it is driven by an intense IR field. Since the infrared laser field strongly dresses the excited-state manifold of the atom and leads to pronounced broadening of resonance lines due to a strong coupling to the continuum[29,30], the excited states can be approximated by a continuum of states. As such, all the incoming XUV harmonics can excite the atom and populate this continuum of states. Our findings reveal the strong-field limit of a range of phenomena observed in attosecond transient absorption experiments at IR intensities $<10^{13}$ W cm$^{-2}$, such as light-induced states[31] and AC Stark shifts[32].

## Discussion

In this paper, we have established a spectroscopic scheme in attosecond science. An independent control over both ionization and strong-field propagation enables us to selectively initiate an electron wavepacket in an extreme strong-field environment and reconstruct its temporal evolution following several coupled quantum paths. The precise attosecond dynamics of the wavepacket and its corresponding hole are encoded in the new, background-free, up-converted XUV frequencies and can be extracted using an interferometric method. The highly anharmonic response of the new harmonics to the XUV-IR delay scan is a clear sign of photo-ionization by photons carrying significantly lower energy than the helium ionization threshold; electrons are liberated by OBE rather than tunneling. These spectroscopic measurements provide hitherto inaccessible

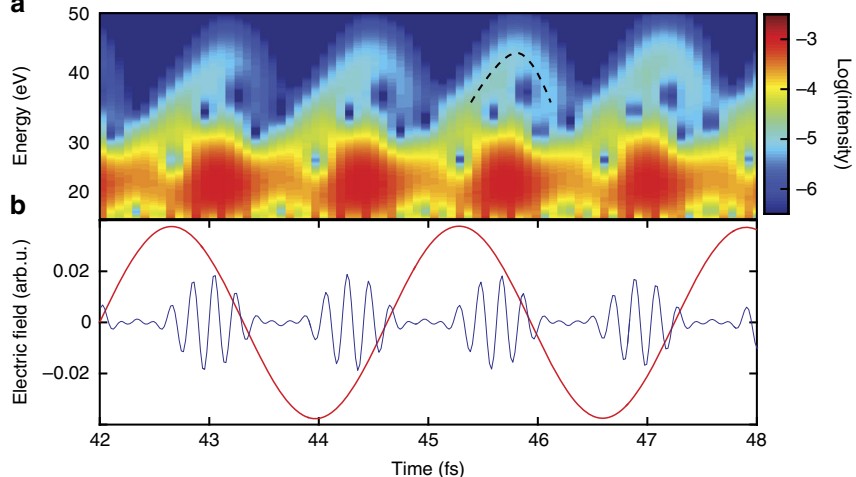

**Fig. 6** Time–frequency analysis of the Schrödinger equation simulation. **a** Gabor time–frequency distribution as a function of time and photon energy for $\Delta t$ = −1575 as (−0.6 IR cycles), presented in logarithmic scale. At high energies >30 eV, the arch-like structures indicate that XUV emission occurs predominantly as a result of strong-field acceleration. Qualitatively, we find agreement of the emission time from the analysis with recombination times of a classical three-step model (illustrated by the dashed black curve). **b** Input IR and XUV fields in arbitrary units

dynamic information on strongly driven systems on a sub-optical-cycle time scale. Coherent nonlinear spectroscopic methods involving frequency up-conversion in the optical and IR regime are at the heart of vibrational and rotational spectroscopy. Applying background-free frequency up-conversion in the XUV regime can significantly advance electronic spectroscopy.

Looking forward, combining the accuracy provided by the self-probing approach with the versatility provided by attosecond photo-ionization in XUV-initiated HHG opens up exciting directions in ultrafast science. While our work studies XUV-initiated HHG in a simple system, the helium atom, we believe that the concept is general and can be applied in future research to probe more complex systems, such as the study of hole dynamics in inner shells[33,34]. The main limitation of the approach lies in the competition of XUV photo-ionization with tunneling ionization, which can create a significant background HHG signal; however, our spectroscopic method can be readily applied to atomic and molecular ions due to their higher ionization potential. The unique combination of attosecond resolution with Ångström precision, provided by the recollision process, will resolve the structural information in such systems and will shed light on fundamental phenomena that are at the heart of attosecond science.

## Methods

**Experimental setup**. Figure 5 shows a schematic description of the experimental setup for XUV-initiated HHG. An amplified Ti:Sapphire laser system operated at 1 KHz repetition rate delivers ~ 23 fs pulses at a central wavelength of 787 nm. Focusing the beam into a continuous flow source gas cell (xenon, ~ 12 Torr) generates an APT. We spatially separate the co-propagating IR and APT beams with the help of a thin aluminum filter (200 nm thickness). Both beams are then refocused by a curved two-segment mirror (600 mm focal length) into the continuous flow target gas cell (helium, ~ 20 Torr) in order to produce XUV-initiated HHG. The inner part of the focusing mirror reflects the APT in the spectral range of 17–27 eV. A piezo stage controls the relative delay of IR and APT with 60 as accuracy. The IR intensity at the target gas cell is adjusted with the help of a motorized iris. The generated harmonic radiation is spectrally resolved by a flat-field aberration-corrected concave grating and recorded by a micro-channel plate detector. The spectrum is imaged by a CCD camera. We integrated over each harmonic on the detector in order to obtain its intensity and subtracted the background that is insensitive to the XUV-IR delay (see Supplementary Note 6 and Supplementary Fig. 10).

For the XUV-IR delay scan experiment (Fig. 2), the local laser intensity was 8.7 ± 1.9 × 10^13 and 7.8 ± 1.9 × 10^13 W cm^−2 in the source and target gas cell, respectively (see Supplementary Note 3). The Fourier transformation of the XUV-IR delay dependent signal includes five oscillation cycles after subtracting the

background. The errors of the Fourier amplitudes and phases represent the obtained standard deviation when varying the offset of the transformation window by ±2 oscillation cycles.

For the SH perturbative measurements (see Supplementary Note 4 and Supplementary Fig. 7), we used ~ 32 fs pulses in a different configuration of the laser system and generated weak SH radiation by propagating the beam prior to focusing through a Type-I barium boron oxide crystal (100 μm thickness). Due to the low conversion ratio (below 5 and 0.5% in the measurements of Fig. 3b and Fig. 3d, respectively), the chirp and the spectrum of the SH can be regarded as locked to the IR. A combination of calcite plates and a half-wave plate ensures group delay compensation and common polarization state for the co-propagating IR and SH beams. We control the SH-IR delay with the help of a fused silica wedge pair. We inserted a fused silica window between source and target gas cells in order to allow synchronization either in the source or in the gas cell; the window has a small hole to allow propagation of the APT. For the perturbative measurements, the peak IR intensities in the source and target gas cells were 7.1 ± 1.9 × 10^13 and 7 ± 3 × 10^13 W cm^−2, respectively.

We estimated the spectral phase of the APT from well-known attochirp values of HHG in xenon gas and the subsequent propagation in the experimental setup; the APT is close to its transform limit (see Supplementary Fig. 11). The perturbative measurement with SH field synchronized with the IR in the source gas cell observed the slope of even harmonics, as predicted by the three-step model (see Fig. 3d). This also provides strong evidence that indeed an APT is generated[21].

**Coulomb-corrected three-step model**. The CCTSM describes XUV-initiated HHG in three steps (see Fig. 1b and Supplementary Fig. 1 for an illustration): photo-ionization by the XUV field via OBE, propagation of the electron in the combined laser-Coulomb potential, and recombination into the ground state. Starting from a strong-field approximation approach[26,33,35], we generalize the model to include OBE and the influence of the atomic potential[25]. OBE essentially enables photo-ionization by XUV absorption and subsequent electron emission over the potential barrier lowered by the IR field. In addition, the excited states under the influence of the strong IR field are treated as a quasi-continuum, motivated by theoretical predictions[29,30]. These modifications are essential since we experimentally observe a strong contribution of the full spectral bandwidth of the APT to the photo-ionization—although H11–H15 are not able to directly ionize helium by themselves when ignoring the effects of the IR field and the Coulomb potential. Although we simplify the interplay of XUV (driving transitions to a quasi-continuum) and IR (creation of a potential barrier that can be passed by an electron without tunneling), the model captures the essential features of the experiment. For a detailed derivation and description, we refer the reader to Supplementary Note 1. Here we only present the essential equations and definitions in atomic units (a.u.).

In the CCTSM, the frequency dipole $d$ along the reaction coordinate $z$ defined by the polarization axis of the laser field is given by

$$d(N\omega_{IR}) \propto \sum_n \sum_{j=1}^M a_n \exp(-i\phi_n) \left( \frac{\pi}{\eta + \frac{i}{2}\left(t_1^{(n,j)} - t_0^{(n,j)}\right)} \right)^{3/2}$$
$$\exp\left[ i\Theta_{XUV-,OBE}\left(t_1^{(n,j)}, t_0^{(n,j)}\right) \right]. \qquad (3)$$

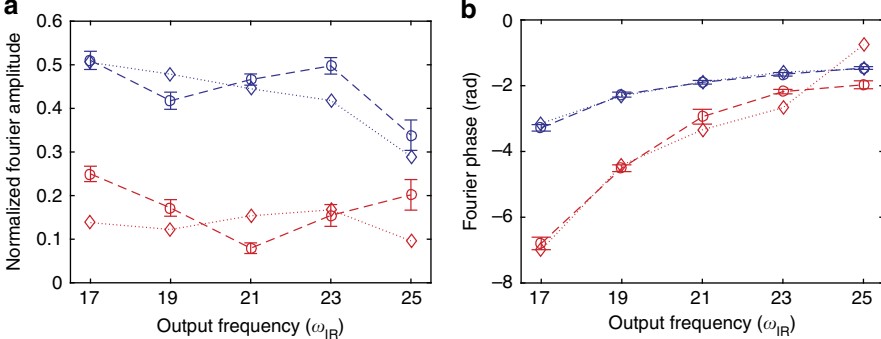

**Fig. 7** Comparison of experimental and reconstructed Fourier data. **a** Normalized Fourier amplitudes as a function of output harmonic frequency and Fourier frequency (circles and dashed lines: experiment, diamonds and dotted lines: theory reconstruction, error bars on experimental data: standard deviation). **b** Like **a**, but showing the Fourier phase

Here $a_n$ is defined as the amplitude of the ionization channel associated with harmonic $n\omega_{IR}$ and $\phi_n$ is the corresponding phase—the object of the reconstruction (see below). Several quantum paths contribute to $d(N\omega_{IR})$, expressed by the two sums over all ionization channels corresponding to input harmonics of frequency $n\omega_{IR}$ and different trajectory classes, numbered with index $j$. This defines the multiple-path quantum interferometer (see Eq. 1). In accordance with the experiment, we take only short trajectories into account; these are subdivided into downhill (extra kick away from nucleus) and uphill trajectories (extra kick towards the nucleus)[18,25], so $M = 2$. The term in large brackets describes quantum diffusion[36], i.e., the decrease of overlap of ground state and emitted wavepacket as it undergoes transversal spread between the times of ionization $t_0^{(n,j)}$ and recollision $t_1^{(n,j)}$. Quantum diffusion leads to a suppression of uphill trajectories since the electron spends a longer time in the continuum compared to downhill trajectories. $\eta \ll 1$ is a small regularization parameter. The quasi-classical action $\Theta_{XUV-,OBE}(t_1^{(n,j)}, t_0^{(n,j)})$ describes the quantum evolution of the system in terms of phase and is given by

$$\Theta_{XUV-,OBE}(t, t') = -n\omega_{IR}t' + n\omega_{IR}\Delta t - S_{OBE,cl}(t, t') - I_p(t - t') + N\omega_{IR}t, \quad (4)$$

with the ionization time $t'$, the recollision time $t$, the classical action along the corresponding trajectory $S_{OBE,cl}(t,t')$ and the ionization potential $I_p = 24.59$eV. The physical meaning corresponds to the three-step picture, with photon absorption ($-n\omega_{IR}t'$) and its XUV-IR delay dependence ($+n\omega_{IR}\Delta t$), electron propagation ($S_{OBE,cl}(t,t')$), and ground state evolution ($-I_p(t-t')$), and finally photon emission ($+N\omega_{IR}t$). A crucial feature of our model is the classical action. It takes the Coulomb force into account and hence must be determined numerically[25]. We use a Runge-Kutta scheme to solve the equations of motion in the combined laser-Coulomb potential. The helium potential is given by a softcore potential[37]

$$V(z) = -\frac{Z_{eff}}{\sqrt{a^2 + z^2}}, \quad (5)$$

with $Z_{eff} = 1.353$ and $a^2 = 0.001$. We explicitly allow for OBE and exclude tunneling. Excitation to states below the ionization threshold is treated effectively as ionization to a continuum.

The APT and the IR field are treated as infinitely extended in time, a sufficiently good approximation for 25 fs IR pulses. The IR intensity used for the reconstruction is $8 \times 10^{13}$ W cm$^{-2}$, and the wavelength is 787 nm. Supplementary Fig. 2 displays ionization and recollision times calculated for these parameters. In the calculations of the SH perturbative experiments, the SH-IR field ratio $\varepsilon$ was taken as 0.05 and the IR intensity as $7.5 \times 10^{13}$ W cm$^{-2}$.

**Integration of the time-dependent Schrödinger equation**. We performed numerical calculations in order to solve the non-relativistic two-electron TDSE for the helium atom (for more details see Supplementary Note 2). Macroscopic phase matching effects have been omitted. We employed the TD-GAS-CI framework[27,38,39], whose accuracy has recently been demonstrated by simulating ionization dynamics in diatomic molecules[28]. In brief, the many-particle wave function is expanded into a selection of time-independent Slater determinants via the GAS concept. The expansion allows for approximations based on physical arguments; in our case, we ignore double ionization. For the GAS division, we use a complete-active-space (CAS) with single excitations out of the active space[27,28] (see Supplementary Fig. 3). It contains a variable amount of orbitals $\nu$ and is denoted CAS$^*$(2,$\nu$). The more orbitals are included in the CAS, the more electronic correlations effect are taken into account and the better the accuracy of the simulation. In particular, ionization from excited states is then possible. We found good convergence with a CAS$^*$(2,12), where orbitals up to the third shell with a maximum magnetic quantum number $m_{max} = 1$ are included.

In order to model our experiment, we used a sine-square IR pulse with a peak intensity of $8 \times 10^{13}$ W cm$^{-2}$ and a full-width half-maximum (FWHM) intensity duration of 32.5 fs (see Supplementary Fig. 4 and Supplementary Table 1). The XUV field comprising harmonics $n = \{11,13,15,17\}$ is given by normalized amplitudes $b_n = \{0.2214,0.4227,0.3127,0.0432\}$ and flat phase and exhibits a $sin^4$-envelope with a FWHM duration of 15.12 fs. Its peak intensity is $5 \times 10^{10}$ W cm$^{-2}$, low enough to suppress two-photon effects. Compared to HHG driven by the IR only, the XUV-initiated HHG creates up to seven orders of magnitude stronger signal (see Supplementary Fig. 5). Scanning the XUV-IR delay leads to a signal similar to the experiment (see Supplementary Fig. 6).

We performed a time–frequency analysis of the dipole acceleration using a short-time Fourier (Gabor) transformation. Figure 6 shows the results of the analysis, plotting the attosecond emission as a function of time and photon energy for $\Delta t = -1575$ as (−0.6 IR cycles). Arch-like structures appear above photon energies of about 30 eV. These structures can be identified with recombination times of the classical three-step model. The arch contains two branches, we find both short trajectories (left branch) and long trajectories (right branch). The prominence of long trajectories in the TDSE results indicates that a quantitative comparison of TDSE and experiment is strongly limited. In our experiment, macroscopic pulse propagation effects favor the short trajectories.

**Reconstruction of the build-up of the electron wavepacket**. The build-up of the XUV-initiated electron wavepacket is characterized by the complex ionization amplitudes $a_n \exp(-i\phi_n)$, where $n$ denotes a quantum path associated with an input harmonic $n\omega$. It is given by

$$a_n \exp(-i\phi_n) = b_n \, exp(-i\beta_n) \times d_{ion}, \quad (6)$$

and contains both relative amplitude $b_n$ and phase $\beta_n$ of the XUV field and the (complex) dipole matrix element $d_{ion}$ associated with the transition from the ground state to an effectively ionized excited state. Such reconstruction enables us to resolve the spectro-temporal properties of the XUV-initiated electron wavefunction.

Our measurement represents a multiple path quantum interferometer, controlled by the XUV-IR delay $\Delta t$. Using the CCTSM we can link the experimental results—Fourier amplitudes and phases—to the amplitudes $a_n exp(-i\phi_n)$, where each amplitude corresponds to an absorption channel initiated by a given input harmonic $n$. We employ a least-square fitting procedure, minimizing the deviation between experimental and theoretical Fourier phases and amplitudes, the latter normalized to the DC amplitude. Since the reconstruction yields relative amplitudes and phases for the wavepacket, we fix the phase of the H13 ionization channel to be $\phi_{13} = 0$ and its amplitude $a_{13}$ to unity, without any associated uncertainty. The calculation does not provide a trajectory from $n = 17$ to $N = 25$; the absence of this trajectory is associated with the error in determining the IR intensity. Although partially visible, the $6\omega_{IR}$ Fourier component does not reach above the noise level compared to the $2\omega_{IR}$ and $4\omega_{IR}$ components (see Fig. 2). Therefore we decided to exclude the $6\omega_{IR}$ Fourier data from the reconstruction procedure; the remaining data are sufficient for the least-square fit. The strong $6\omega_{IR}$ component of H19 has a different origin; it is due to interference between the XUV-initiated harmonic light and the background harmonic from the ionizing APT[40].

We obtain a good agreement between experiment and theory as evidenced by the error bars (standard deviation) in Fig. 4a, b. Figure 7 shows a comparison of experimental Fourier amplitudes and phases with the reconstructed values. $6\omega_{IR}$ oscillations are also obtained from the reconstruction (not shown), but their amplitudes are not able to overcome the experimental noise floor.

**Data availability**. The data that support the findings of this study are available from the corresponding author on request.

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

## Acknowledgements

We thank Yann Mairesse, Mette B. Gaarde, Kenneth J. Schafer, Albert Stolow, David J. Tannor, and Andreas Kaldun for helpful discussions. N.D. is the incumbent of the Robin Chemers Neustein Professorial Chair. N.D. acknowledges the Minerva Foundation, the Israeli Science Foundation, the Crown Center of Photonics, and the European Research Council for financial support. M.K. acknowledges financial support by the Minerva Foundation and the Koshland Foundation. H.R.L. acknowledges financial support by the DAAD (German Academic Exchange Service), the Studienstiftung des Deutschen Volkes, and the Fonds der Chemischen Industrie. S.B. acknowledges financial support from the BMBF (Federal Ministry of Education and Research, Germany) in the frame of the Verbundprojekt FSP 302.

## Author Contributions

N.D. conceived and supervised the study. D.A. designed the experimental setup. D.A. and M.K. built the experimental setup, performed the experiment, and analyzed the data. M.K., G.O. and D.A. devised the theory model. B.D.B. supported the operation of the laser system. H.R.L. and S.B. conceived and performed the time-dependent Schrödinger equation calculations. All authors discussed the results and contributed to the final manuscript.

## Additional information

**Competing interests:** The authors declare no competing financial interests.

