## [Peer Review File · Nature Communications]

Reviewers' comments:

Reviewer #2 (Remarks to the Author):

In the revised manuscript the authors have changed the work's perspective: the multi-dimensional spectroscopy approach has been replaced by self-probing spectroscopy. Now the title reflects the main objective of the manuscript. The experimental method is properly explained and it is applied to the reconstruction of the electron wavepacket generated by the XUV pulse in the presence of a strong infrared field. What is effectively measured is the spectral amplitude and phase at four frequencies, corresponding to four harmonic peaks. The result is for sure interesting but, at least this is my impression, it does not demonstrate that the proposed experimental technique can really "open up new and exciting directions in ultrafast science". It represents a proof-of-principle application to a relatively simple case. The reported results do not justify authors' claim about the possibility to obtain a real-time probing of complex multi-electron phenomena in molecular and solid state physics.

Reviewer #4 (Remarks to the Author):

The manuscript by Doron et al. "Self-probing spectroscopy of XUV photo-ionization dynamics in a strong-field environment" demonstrates how the liberated electron can perform a self-probing measurement. In the manuscript, the authors claimed that they have developed a conceptually new spectroscopic approach called 'self-probing spectroscopy'. The author applied the technique to analyze the high harmonic radiation obtained with the XUV and IR laser pulses. It is demonstrated that the ultraviolet attosecond pulses generated in the first target initiate an electron wavepacket in the second target by photo-ionization. Then, the liberated electron is driven by the strong IR pulse, and recombine to the parent ion. It finally emits the attosecond pulse in the second target. By applying a weak perturbative field, the authors studied either the XUV excitation step or the strong-field interaction. The superposition of the weak second harmonic field imposes an asymmetry in the electron recollision dynamics. The modulation of the harmonic intensity is observed as a function of the time delay. These measurements identify the dynamics of XUV-initiated HHG. Also, the amplitude and phase of the electron wave packet generated by the XUV excitation is reconstructed.

While I am reading the manuscript, I could not find the difference between 'self-probing spectroscopy' and conventional high harmonic spectroscopy. The explanation (line 33 – 47 in page 2) of the self-probing spectroscopy is exactly the same with conventional high harmonic spectroscopy where the remaining part of the electronic wavefunction plays a role of the reference or the signal to be measured. The technique that they applied in the manuscript (use of the perturbing field) is also very similar with their previous works (For example, Dudovich, N. et al. Nature Phys. 2, 781-786 (2006)). Therefore, I do not think their approach is conceptually (and completely) new spectroscopic technique. Therefore, I cannot recommend the manuscript for the publication in Nature Communication.

I also have some comments below.

1) The author mentioned in the abstract, "a strong infrared field manipulates its coherence". However, the photoionization and recollision process discussed in the manuscript are all coherent processes. I wonder how the coherence can be controlled. There is no explanation about the coherence in the manuscript.

2) I wonder if the technique applied in the manuscript is sensitive to the phase in the photoionization (i.e. Wigner group delay). It would be nice if the authors can comment.

Reviewer #2

In the revised manuscript the authors have changed the work's perspective: the multi-dimensional spectroscopy approach has been replaced by self-probing spectroscopy. Now the title reflects the main objective of the manuscript. The experimental method is properly explained and it is applied to the reconstruction of the electron wavepacket generated by the XUV pulse in the presence of a strong infrared field. What is effectively measured is the spectral amplitude and phase at four frequencies, corresponding to four harmonic peaks. The result is for sure interesting but, at least this is my impression, it does not demonstrate that the proposed experimental technique can really “open up new and exciting directions in ultrafast science”. It represents a proof-of-principle application to a relatively simple case. The reported results do not justify authors' claim about the possibility to obtain a real-time probing of complex multi-electron phenomena in molecular and solid state physics.

In our paper we introduce a new approach for attosecond spectroscopic measurements which integrates the two main branches in attosecond science – photoelectron spectroscopy and HHG (or self-probing) spectroscopy. We agree with you that our approach is demonstrated in a simple system, the helium atom. The simplicity of the system enables us to establish the new approach and study its internal dynamics.

Our experiment allows us to study a new regime which has not been explored so far – XUV-induced photo-ionization of an atomic system modified by the presence of a strong infrared laser field ($> 5 \times 10^{13} \text{ W/cm}^2$). We demonstrate how this measurement enables the reconstruction of the dynamics involved in the photo-ionization process. Although the photon energy of three out of the four harmonics is insufficient for ionization, we find that the ionization bandwidth in the IR-driven helium atom reaches as deep as 9 eV below the atom's field-free ionization potential. Surprisingly, we find that while these quantum paths are initiated substantially below the ionization threshold, they follow almost purely classical

recollision trajectories under the influence of the strong laser field. Our findings provide the first experimental study of this regime and are in agreement with previous theoretical predictions (see Refs. 29, 30).

While we believe that this measurement opens a new and exciting direction in attosecond spectroscopy, we accept your criticism and toned down our claims. Following your comment we have modified some of the statements in the abstract and the conclusion part of the paper.

- a. Abstract – we have removed the sentence “These observations bear the prospect of real-time probing of complex multi-electron phenomena that are at the forefront of attosecond science.”
- b. We have completely re-written the conclusion part by replacing the last paragraph, originally aimed at future perspectives, with the following paragraph (lines 276-288):

“Looking forward, combining the accuracy provided by the self-probing approach with the versatility provided by attosecond photo-ionization in XUV-initiated HHG, opens up new and exciting directions in ultrafast science. **While our work studies XUV-initiated HHG in a simple system, the helium atom, we believe that the concept is general and can be applied in future research to probe more complex systems. Possible directions include the study of hole dynamics in inner shells [Leeuwenburgh 2013, Brown 2016] or photo-excitation in the condensed phase [Ghimire2011, Vampa2015]. The main limitation of the approach lies in the competition of XUV photo-ionization with tunneling ionization which can create a significant background HHG signal; however, our spectroscopic method can be readily applied to atomic and molecular ions due to their higher ionization potential. The unique combination of attosecond resolution with Angstrom precision, provided by the recollision process, holds the potential of resolving the structural information in such systems, and will shed new light on fundamental phenomena that are at the heart of attosecond science.**”

Reviewer #4

1. In your review you raise a serious concern about the novelty of our work with respect to conventional HHG spectroscopy.

While I am reading the manuscript, I could not find the difference between ‘self-probing spectroscopy’ and conventional high harmonic spectroscopy. The explanation (line 33 – 47 in page 2) of the self-probing spectroscopy is exactly the same with conventional high harmonic spectroscopy where the remaining part of the electronic wavefunction plays a role of the reference or the signal to be measured.

We apologize for creating a misunderstanding here. Indeed, the self-probing scheme is a well-established scheme, being *identical* to HHG spectroscopy; both definitions are used interchangeable. In lines 33-47 we describe a review of previous studies based on the self-probing approach. The self-probing (or HHG spectroscopy) approach, depicted schematically in Fig. 1a, has been extremely successful in resolving the evolution of electronic and nuclear wavefunctions. Our approach – photo-ionization self-probing spectroscopy – lifts main limitations of conventional self-probing (or HHG spectroscopy). The limitations originate from the first step of HHG – IR-field-induced tunnel ionization. Since the tunneling probability decays exponentially with electron binding energy, it allows probing of valence shell orbitals only; inner shells are out of reach. Moreover, in systems with multiple orbitals, the tunneling mechanism dictates the relative amplitudes and phases of the associated ionization channels. Finally, tunneling is temporally constrained to a narrow time window around the peak of the optical field.

In our paper we demonstrate how *the self-probing approach can be initiated with XUV photo-ionization*. Linear photo-ionization using XUV attosecond pulses, in contrast to field-induced tunneling, enables us to control the timing – by controlling the relative XUV-IR delay – and to address arbitrary electronic states in the matter under scrutiny – by engineering the ionizing XUV spectrum. While not being at the focus of our work, we show an example of engineering the ionizing XUV spectrum using a SH-IR field in the source cell; controlling their delay we accurately manipulate the XUV spectrum and therefore, directly, the ionization mechanism itself (see Fig. 3d and e).

Our study reveals several fundamental differences between conventional HHG and XUV-initiated HHG:

- a. In conventional HHG both field-induced tunneling ionization and recollision are induced by the same strong IR laser field and therefore *cannot be independently controlled*. This coupling is one of the primary limitations in self-probing / HHG spectroscopy. In contrast, XUV-initiated HHG enables the *decoupling* of these two degrees of freedom and their independent control. Indeed, in Fig. 3, we demonstrate that these two degrees of freedom are decoupled and reveal their underlying dynamics.
- b. In conventional HHG there is a direct mapping between each moment of ionization to a moment of recollision and emitted harmonic energy. In contrast, in XUV-initiated HHG the picture is more complex. Here the XUV field initiates several independent channels of ionization. Strong-field recollision trajectories superimpose them coherently into each individual emitted harmonic. Such a coherent superposition, together with accurate control over the XUV-IR delay, enables us to resolve the phase associated with each ionization channel and extract its complex contribution. We demonstrate this important aspect of the measurement by reconstructing the electronic wavefunction at the moment of ionization (see Fig. 4).

In order to avoid any misunderstanding we have modified the manuscript as following:

- a. In the abstract, we changed one sentence in order to guide the reader's attention to the fact that photo-ionization in the presence of a strong laser field is at the center of our study:
“Our measurements resolve the internal clock provided by the self-probing mechanism, **obtaining a direct insight into the build-up of photo-ionization in the presence of the strong laser field.**”
- b. In the second paragraph of the manuscript (line 33, p. 2) we have clarified the identity of the self-probing approach and HHG spectroscopy:
“This spectroscopic approach, known as attosecond self-probing **or HHG spectroscopy**, exploits a built-in pump-probe process driven by an intense IR laser field every half-cycle of its oscillation [Kim 2014].”
- c. In lines 54-56, p. 3, we have modified the first sentence to be:
“In this paper we experimentally demonstrate a conceptually new spectroscopic approach in attosecond science that integrates one of the most fundamental light-matter interactions - single-photon ionization - **with self-probing spectroscopy.**”

We hope that these modifications remove the misunderstanding and clarify the conceptual difference between conventional HHG / self-probing spectroscopy, and our work.

2. In your review you refer to the perturbative approach applied in our paper:

The technique that they applied in the manuscript (use of the perturbing field) is also very similar with their previous works (For example, Dudovich, N. et al. Nature Phys. 2, 781-786 (2006)).

Indeed, the two-color scheme has been applied to study the underlying dynamics in conventional HHG. In the current paper we apply this approach and study, for the first time, the underlying dynamics in *XUV-initiated HHG*. The perturbative measurement leads us to the following conclusions:

- a. Field-induced tunneling does not play a role in the ionization mechanism. Instead, XUV photo-ionization dominates ionization.
- b. While the electron is ionized well below the ionization threshold, they follow a classical recollision trajectory as it interacts with the laser field.

Accordingly, we have modified the manuscript as following:

In the section “Perturbative study of the mechanism” we added a new sentence that refers to the application of the two-color scheme to study conventional HHG (lines 155-156):

“This approach has been previously applied to probe the internal dynamics of the conventional HHG mechanism [Dudovich 2006].”

3. *The author mentioned in the abstract, “a strong infrared field manipulates its coherence”. However, the photo-ionization and recollision process discussed in the manuscript are all coherent processes. I wonder how the coherence can be controlled. There is no explanation about the coherence in the manuscript.*

We completely agree with your comment. In the original submitted manuscript to Nature, the focus of the work was on a multi-dimensional spectroscopy interpretation. The sentence you are mentioning is a remainder from this interpretation and uses the language of multi-dimensional spectroscopy. In this context, coherences mean the superposition of several quantum states or quantum paths, in our case, the photo-ionization channels. We manipulate the relative phase and amplitude by propagation in the IR field.

This sentence in the abstract now reads (lines 16-19):

“Here, extreme-ultraviolet attosecond pulses initiate an electron wavepacket by photo-ionization, a strong infrared field **controls its motion**, and finally electron-ion collision maps it into re-emission of attosecond radiation bursts.”

4. *I wonder if the technique applied in the manuscript is sensitive to the phase in the photo-ionization (i.e. Wigner group delay). It would be nice if the authors can comment.*

We thank you for raising this excellent comment. Indeed, the Wigner group delay that is associated with the XUV-driven photo-ionization is encoded in the XUV-initiated high harmonic generation mechanism. However, as a group delay it corresponds to the entire electron wavepacket, including all the ionization channels. Since in our experiment all ionization channels start from the same ground state we lack a reference for extracting the group delay.

The Wigner group delay has been measured by comparing photoemission from two or more different initial states. RABBITT or attosecond streaking methods with a weak IR field have been applied to study the photo-ionization process and to resolve relative group delays between different initial states. We believe that XUV-initiated HHG can serve as a new and extremely sensitive approach to resolve such delays.

Following your comment, in future research we intend to study Wigner group delays in XUV-initiated HHG by performing a differential measurement of two atomic species as targets, for example helium and neon. Keeping identical experimental conditions, we will be able to extract the relative group delay between the two atoms with extremely high precision.

In summary, we would like to thank the referees for their careful review and important comments. We hope that our detailed reply and clarifications have addressed the concerns raised by the referees.

Best regards,

Nirit Dudovich (on behalf of all authors)

REVIEWERS' COMMENTS:

Reviewer #2 (Remarks to the Author):

I confirm my last review. The results are interesting and well presented, but do not justify authors' claim about the possibility to obtain a real-time probing of complex multi-electron phenomena in molecular and solid state physics: they are appropriate for a more technical journal.

Reviewer #4 (Remarks to the Author):

I am very satisfied with the revised manuscript. The authors replied to all my comments, and modified the manuscript properly. Originally, I initially made a negative opinion, but I would like recommend the revised manuscript for publication in Nature Communications.

Reviewer #2

I confirm my last review. The results are interesting and well presented, but do not justify authors' claim about the possibility to obtain a real-time probing of complex multi-electron phenomena in molecular and solid state physics: they are appropriate for a more technical journal.

We would like to thank you for the interest you find in our work. In the current version of the manuscript as a future prospective, we decided to remove the prospects for solid state spectroscopy and to focus on the potential of XUV-initiated HHG as a spectroscopic technique for probing inner-shell electron dynamics. This research direction is a natural continuation of our present work; it will exploit the full flexibility of the technique by controlling the photo-ionization energy in addition to the temporal degree of freedom, demonstrated in our work. The technical challenges in these experiments are the necessary high XUV photon flux of the ionization beam in order to ionize the inner shell electrons, and the competition between standard HHG from valence electrons and XUV-initiated HHG from inner shell electrons. Our group and other groups have developed HHG sources which can provide significantly higher flux than conventional generation schemes, for example very high pressure gas cells or generation through gas filled capillary. Moreover, the expected emission energy range by XUV-initiated HHG from inner shell electrons will be much higher than the standard HHG from valence electrons due to the large difference in the ionization potential.

Accordingly, we modified the future prospective. The final paragraph now reads:

“Looking forward, combining the accuracy provided by the self-probing approach with the versatility provided by attosecond photo-ionization in XUV-initiated HHG,

opens up exciting directions in ultrafast science. While our work studies XUV-initiated HHG in a simple system, the helium atom, we believe that the concept is general and can be applied in future research to probe more complex systems such as the study of hole dynamics in inner shells [Leeuwenburgh2013, Brown2016]. The main limitation of the approach lies in the competition of XUV photo-ionization with tunneling ionization which can create a significant background HHG signal; however, our spectroscopic method can be readily applied to atomic and molecular ions due to their higher ionization potential. The unique combination of attosecond resolution with Angstrom precision, provided by the recollision process, will resolve the structural information in such systems, and will shed light on fundamental phenomena that are at the heart of attosecond science.”

Reviewer #4

I am very satisfied with the revised manuscript. The authors replied to all my comments, and modified the manuscript properly. Originally, I initially made a negative opinion, but I would like recommend the revised manuscript for publication in Nature Communications.

We would like to thank you for your positive feedback and for the recommendation to publish our work.

Best regards,

Nirit Dudovich (on behalf of all authors)